# 🐯 TP-Eval: Tap Multimodal LLMs' Potential in Evaluation by Customizing Prompts

## Abstract

Recently, multimodal large language models (MLLMs) have received much attention for their impressive capabilities. The evaluation of MLLMs is becoming critical to analyzing attributes of MLLMs and providing valuable insights. However, current benchmarks overlook the problem of prompt sensitivity - minor prompt variations may lead to significant performance fluctuations. Thus, inappropriate prompts may obscure the models' capabilities, underestimating the models' performance. Moreover, different models have different preferences for different prompts, and thus, using the same prompt for all models will cause evaluation bias. This paper analyzes this deficiency in existing benchmarks and further introduces a new evaluation framework named TP-Eval, which introduces a prompt customization method to reduce evaluation biases and tap models' potential. TP-Eval will rewrite the original prompts to different customized prompts for different models. In particular, we propose some well-designed modules for prompt customization tailored to the scenario of MLLM evaluation. Extensive experiments demonstrate the effectiveness of our approach to uncovering models' capabilities, and TP-Eval should benefit the community in developing more comprehensive and convincing MLLM evaluation benchmarks.

## 1 Introduction

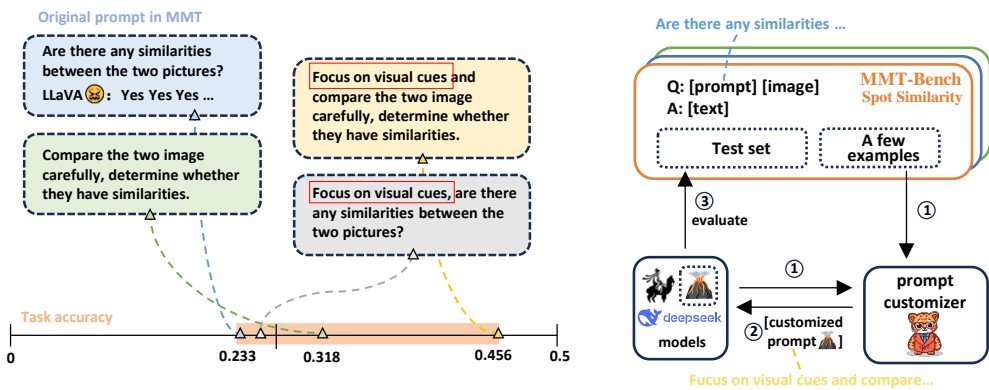

(a) Example of prompt sensitivity in multi-modal benchmark.      (b) Framework of TP-Eval.

Figure 1: (a) shows underestimation caused by unsuitable prompts in MMT-Bench, (b) shows our proposed evaluation framework resolving this by customizing prompts.

Large language models (LLMs), such as ChatGPT, and Claude, are becoming a milestone in achieving artificial general intelligence (AGI). Recently, beyond text conversation, multimodal large language models (MLLMs), like GPT-4o (Achiam et al. (2023)), Deepseek (Lu et al. (2024)), InternVL (Chen et al. (2024)) and LLaVA (Liu et al. (2024a)), have received much attention for their im-

pressive capabilities to understand multimodal inputs (this paper focuses on image and text). Subsequently, researchers present various benchmarks to evaluate their performance in different scenarios. Most apply prompt-based benchmarking approaches to ask models multimodal questions and assess their responses. For instance, MMT-Bench by Ying et al. (2024) comprehensively evaluates performance in 162 general tasks spanning 32 categories. Meanwhile, MMMU by Yue et al. (2024) encompasses six core disciplines drawn from university curricula and assesses performance on multidisciplinary tasks requiring domain-specific knowledge and meticulous reasoning. Convincing benchmarking is crucial to analyze the attributes of models, provide valuable insights, and guide the development of MLLMs.

Nevertheless, recent research (Zhan et al. (2022; 2023; 2024)) found that LLMs and MLLMs exhibit pronounced sensitivity to prompt variations. Thus, minor modifications to questions in benchmarks may lead to significant output differences. This makes prompt-based benchmarking unreliable since models' low accuracy may be owed to unsuitable prompts, not their inner capability. Furthermore, many MLLMs' benchmarks use simple and uniform prompts for all samples in a specific task, which aggravates the problem and causes general underestimation. Additionally, different models show various sensitivity to the same prompt changes, and existing evaluation frameworks fail to consider such prompt-induced bias and may not be able to conduct a convincing comparison.

To address the aforementioned deficiencies, this paper introduces TP-Eval, a novel evaluation framework for MLLMs that customizes optimal prompts for different models to fully tap their potential during evaluation while mitigating the effects leading to performance underestimation by prompt sensitivity. We posit that this framework enables researchers to assess the strengths and weaknesses of various models more accurately. To ensure fairness across models while also managing labor costs, it is essential for the prompt customization process to be automated. A relevant technique is automatic prompt optimization, as exemplified by recent methods such as ProTeGi Pryzant et al. (2023) and OPRO Yang et al. (2023), which employ an optimizer-scorer architecture. These methods generate multiple candidate prompts and score them on a training set to identify the most effective option.

Inspired by this, TP-Eval implements prompt customization through automatic prompt optimization tailored to MLLMs' evaluation. In particular, related prompt optimization methods consider text only, while our prompt customization incorporates text with images. Moreover, the data scale of the MLLM benchmark is usually limited (e.g., 20 validation samples per task in MMT-Bench) due to the high construction cost, while related prompt optimization methods did not consider this few-shot scenario and easily caused overfitting. Thus, our method introduces a novel error introspection from wrong responses and employs some designs to limit the prompt semantic change. They significantly improve the performance of our method.

We conduct extensive experiments to reveal the presence of prompt-induced underestimation and bias in MLLM evaluation and demonstrate that the TP-Eval framework effectively mitigates these issues. The primary contributions of this paper can be outlined as follows:

- We identify and analyze prompt design deficiencies in existing MLLMs' benchmarks that lead to underestimation and evaluation bias due to prompt sensitivity in MLLMs.
- We propose TP-Eval, a novel evaluation framework for MLLMs that customizes optimal prompts for distinct models and makes it practical through automatic prompt optimization tailored to MLLMs' benchmarks.
- We conducted extensive experiments on advanced MLLM benchmarks and various MLLMs to demonstrate the effectiveness of our method in alleviating the underestimation bias in evaluation.

## 2 MULTIMODAL LARGE LANGUAGE MODEL EVALUATION

### 2.1 ANALYSIS FOR EXISTING BENCHMARKS

In order to comprehensively evaluate the overall reasoning capabilities of MLLMs, many benchmarks have been proposed, encompassing a wide range of tasks that assess various aspects of model performance. Some notable benchmarks are MMBench by Liu et al. (2024b), MMMU by Yue et al. (2024), MM-Vet by Yu et al. (2023), SEED-Bench by Li et al. (2023) and MMT-bench by Ying et al.

| Prompt | LLaVA | DeepSeek |
|:---:|:---:|:---:|
| Is the person in the picture wearing a helmet? | 0.65 | 0.79 |
| **Evaluate if** the individual in the picture wearing adequate headgear that provides safety and **visibility** to minimize interpretation ambiguity. | 0.88 | 0.61 |
| **Is** the individual in the picture wearing an adequate headgear that provides safety and **is visible** to minimize interpretation ambiguity? | 0.69 | 0.83 |

Table 1: Similar prompt changes have different effects on two models for helmet anomaly detection task in MMT-Bench.

(2024). Unlike the prompts used in text-only benchmarks for LLMs, MLLMs' benchmarks primarily convey the majority of the question information through images. Additionally, considering the substantial human effort required to design a specific textual prompt for each image, the prevailing approach is to provide a simple prompt template or even an identical prompt for a given task, like `How many {<object>} are there in the image?` for counting task and `What emotion is expressed in the artwork in the picture?` for artwork emotion recognition task.

However, extensive research demonstrates that LLMs are sensitive to minor modifications of textual prompts, so whether MLLMs are also sensitive to prompt design in existing benchmarks? As shown in Fig. 1a, the original prompt `Are there any similarities between the two pictures?` of the spot similarity task in MMT-bench will lead to an anomalous response from the llava-1.5-7b, who answered `Yes` to all 180 questions, resulting in an extremely low accuracy rate. However, by slightly rephrasing the question, the model achieves nearly double accuracy. This suggests that the model's capability is underestimated due to inadequate prompt design. Further investigation into the accuracy change brought from the phase `Focus on visual cues` indicates that the model's responsiveness to prompts is challenging to predict by humans, raising questions about whether seemingly reasonable prompts in existing benchmarks can truly and accurately assess the model's capabilities.

Nevertheless, designing more suitable prompts for all models in benchmarks won't solve this problem fundamentally since different MLLMs' model architecture and training data are different, leading to different behaviors, preferences, and sensitivity to prompts. Previous research on prompt engineering for LLMs has indicated that prompt design strategies effective for one model may prove ineffective for another (Sclar et al. (2023)). Similar phenomena have also been observed in MLLMs. An intuitive example can be found in Table 1 whereby customizing a more detailed prompt for LLaVA will enhance the accuracy of the helmet anomaly detection task in MMT-Bench. However, this specific prompt declined DeepSeek's accuracy significantly. When utilizing this prompt, LLaVA's performance will surpass that of DeepSeek, and subtle adjustments may reverse this outcome, which implies that comparing the outputs of two models under an identical prompt may not necessarily provide a valid performance ranking.

The above discussions regarding prompts indicate that the existing benchmarks and evaluation methods may not accurately assess the true capabilities of models or facilitate a reliable comparison of their performance, and simplistic prompt templates in MLLM benchmarks exacerbate this issue. Action should be taken to mitigate the influence of prompts on model evaluations.

## 2.2 IDEAL EVALUATION

The ideal evaluation should be able to evaluate the true capabilities of the model. However, due to the significant performance influence caused by prompt design, how do we define the true capabilities during evaluation? We argue that models' true capabilities are performance under optimal prompts, considering that users will also refine the prompts to get desirable responses when using MLLMs. The optimal prompts should be derived from slight modifications from the benchmarks' original prompts while maintaining the semantic integrity of the task instructions. The optimal prompts for different models may be identical or different. Therefore, we propose TP-Eval, an evaluation

framework that customizes the best prompts for each model in each task, thereby tapping their potential and uncovering their true capabilities.

Manual exploration of optimal prompts during evaluation is time-consuming and impractical. Inspired by existing works on automatic prompt optimization for LLMs, we propose to use an automated prompt customizer to leverage original prompts from benchmarks and a few examples to customize specific prompts for each MLLM under evaluation, thereby tapping their potential.

However, existing text-only prompt optimization methods are not applicable. On the one hand, the data scale for multi-modal tasks is relatively small, especially for evaluation data, which necessitates that the prompt customizer possesses a strong few-shot capability, which is overlooked by existing methods. On the other hand, the desirable prompt customization requires a new framework to utilize visual information beyond text, and the cost associated with calling MLLM APIs is prohibitively high, making extensive calls impractical. Therefore, a novel prompt customization method tailored specifically for multi-modal benchmarks is needed.

## 3 RELATED WORKS

### 3.1 RESEARCH ON PROMPT SENSITIVITY

Some studies have revealed that even minor prompt modifications, which have negligible impact on human semantic understanding, can lead to significant shifts in the output of LLMs (Zhan et al. (2022; 2023)). This property has been widely exploited in the creation of adversarial examples, where small perturbations to the embeddings or input text can induce the model to generate incorrect or misleading answers (Zhan et al. (2024)). This sensitivity allows minor adjustments to questions in LLM benchmarks to significantly impact the final evaluation performance. Recent research has begun exploring variations in prompt formatting to achieve better results (Sclar et al. (2023)). Similar phenomena also occur for MLLM. However, addressing this deficiency in MLLM benchmark design remains relatively underexplored. In this work, we provide a detailed analysis of prompt design issues and introduce an effective evaluation framework with prompt customization to avoid the above problems and bias from prompts.

### 3.2 PROMPT ENGINEERING & OPTIMIZATION

Prompt engineering seeks to identify effective prompts for LLMs to optimize their task performance. To minimize manual effort, researchers have explored automatic prompt optimization, broadly categorized into continuous and discrete methods. Discrete methods directly optimize natural language prompts using techniques such as reinforcement learning (Zhang et al. (2022)) or prompt editing (Prasad et al. (2022)). In contrast, continuous methods (Lester et al. (2021); Li & Liang (2021)) perform optimization within the LLMs' embedding space, enabling gradient-based approaches. Given the unprecedented capabilities of LLMs, recent research has started leveraging them as prompt optimizers. For example, Yang & Li (2023) integrates LLMs with evolutionary algorithms to enhance prompt optimization, while Yang et al. (2023); Pryzant et al. (2023) focuses on adapting concepts and techniques from gradient-based model optimizers, including gradient descent (Pryzant et al. (2023)) and momentum methods (Yang et al. (2023)), for LLM-based prompt optimization.

Our work follows discrete methods and employs MLLM as prompt optimizers. In particular, we combine error introspection, semantic change, and accuracy as "pseudo-gradients" proposed by Tang et al. (2024) to guide the MLLM optimizer in the multimodal scenario. We also introduce a final re-ranking scheme for better performance.

## 4 METHOD

### 4.1 OVERVIEW

Fig. 1b illustrates the overall pipeline of TP-Eval, given the initial text prompts with a few examples $\mathcal{D}_{few}$ from the evaluation dataset for a task, and a MLLM $\mathcal{M}_T$ to be evaluated. We introduce a prompt customization method to obtain the optimal prompt $p^*$ for $\mathcal{M}_T$, then do an ideal evaluation to maximize its potential on the original test set $\mathcal{D}_{test}$.

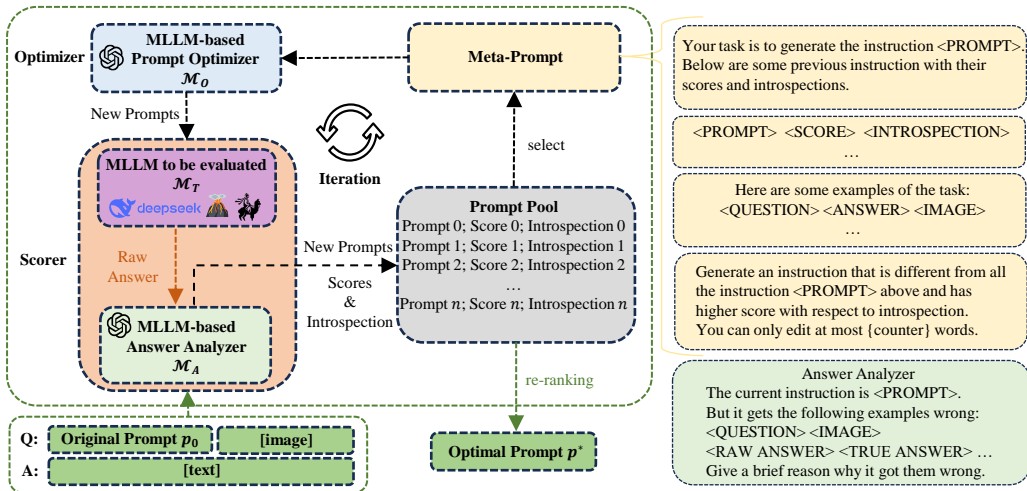

Figure 2: The overview of our automatic prompt customization structure.

We show the overall framework of our customization method in Fig. 2. Starting from the initial text prompt $p_0$ for a task from the multimodal evaluation dataset, we utilize GPT-4o mini as an optimizer $\mathcal{M}_O$ and a few examples $\mathcal{D}_{few}$ (questions and answers) from the evaluation dataset to obtain an optimal prompt $p^*$ for the MLLM $\mathcal{M}_T$. Specifically, we first feed $p_0$ to a scorer, which consists of the $\mathcal{M}_T$ and an answer analyzer $\mathcal{M}_A$ (GPT-4o mini), to output the scores and introspection. Then we use these results to construct a well-designed meta-prompt for the optimizer $\mathcal{M}_O$ to obtain optimized prompts $P_1 = \{p_1, p_2, \cdots, p_n\}$. We feed them to the scorer and iteratively run this framework to collect $N$ sets of optimized prompts $\{P_1, P_2, \cdots, P_N\}$ with their scores. Finally, we select the optimal prompt $p^*$ according to the scores. Please note that we will feed the corresponding images to $\mathcal{M}_T$ and corresponding images and answers to $\mathcal{M}_A$. We will introduce the details of the prompt customization method in the following.

## 4.2 SCORER

In the $i$-th iteration, we feed the prompt set $P_i$ (using $p_0$ in the first iteration) to the scorer to obtain the corresponding scores and introspection (i.e., pseudo gradient) of each prompt.

### 4.2.1 SCORE GENERATION

We first feed these prompts with corresponding images to $\mathcal{M}_T$ to obtain models' responses. Then considering the variations of answers and most benchmarks apply choice questions, we use $\mathcal{M}_A$ (GPT4o-mini) extract choices and then compute the accuracy $a_{p_i}$ on $\mathcal{D}_{few}$ for $p_i$.

Using accuracy as a reward only may lead to drastic changes in the new prompt and destroy the optimization. Thus we utilize a semantic similarity metric as proposed by Tang et al. (2024) to limit the changes in each iteration. Specifically, we use BERT by Kenton & Toutanova (2019) to extract the embedding of the current prompt $p_i$ and the original prompt $p_0$, then calculate their cosine similarity as $s_{p_i}$.

We combine $a_{p_i}$ and $s_{p_i}$ as the final score $c_{p_i} = \alpha a_{p_i} + (1-\alpha)s_{p_i}$, where $\alpha$ is a weighting coefficient to make a trade-off between optimization and original semantic meaning maintain.

### 4.2.2 INTROSPECTION GENERATION

We argue that scores are quantitative and not informative enough, especially in the few-shot examples, and thus, we introduce to employ additional introspection during optimization. Specifically, we aim to help the optimizer better understand the deficiencies in the current prompt. To achieve this, we represent introspection $I_i$ on the incorrect responses in $\mathcal{D}_{few}$ of $\mathcal{M}_T$ under $p_i$, allowing $\mathcal{M}_O$ to

explicitly reference the reasons for these errors when generating new prompts. We show the prompt structure to generate introspection in Fig. 2 and the full prompt in the supplementary materials.

### 4.3 OPTIMIZER

We use the optimizer $\mathcal{M}_O$ (GPT4o-mini) to generate a new prompt set $P_{i+1}$ from all history prompts $\{P_0, \cdots, P_i\}$. Specifically, we design a meta-prompt as shown in Fig. 2 and complete it using $K$ prompts with Top-$K$ scores from $\{P_0, \cdots, P_i\}$. We also feed their scores and introspection to the optimizer. The meta prompt is composed of four parts: description, pseudo gradient (i.e., prompts with their scores and introspection), examples (questions with ground-truth answers from $\mathcal{D}_{few}$), and instruction. The description is used to describe the prompt optimization tasks. The pseudo gradient and examples are used to provide information for the optimization. The instruction is used to generate new prompts. In particular, to ensure smooth optimization and not overlook optimal prompts, we use a decaying edit distance `You can only edit at most {counter} words` to limit the changes. Please note that for identical question benchmarks (e.g., MMMU), we will add an initialized meaningless phrase and optimize it rather than the whole prompt, see the experiments section for more details.

### 4.4 ITERATION

We use the above scorer-optimizer framework iterative to obtain $N$ prompt set $\{P_1, P_2, \cdots, P_N\}$ with scores for each prompt. Then we select the optimal prompt from all history prompts.

In contrast to related prompt optimization methods using large-scale training data to obtain candidate prompts and selecting the optimal prompt with the highest accuracy, MLLM evaluation can provide only limited examples in the optimization. Thus, we have to consider the problem of overfitting and bias in the few examples. The introduced semantic cosine similarity and decaying edit distance can alleviate this problem. Moreover, in the selection of the optimal prompts, we employ a higher weighting coefficient $\alpha^* > \alpha$ to re-compute each prompt's score and select the prompt with the highest score.

## 5 EXPERIMENT

### 5.1 EXPERIMENTAL SETUP

**Models.** The MLLMs to be evaluated (i.e., $\mathcal{M}_T$) are `LLaVA-1.5-7B`, `DeepSeek-VL-7B`, `Mini-InternVL-Chat-4B-V1-5`. We use `GPT-4o-mini` for optimizer ($\mathcal{M}_O$) and answer analyzer ($\mathcal{M}_A$).

**Benchmarks.** We use MMT-Bench and MMMU as the evaluation benchmarks. MMT-Bench is designed for the evaluation of general capabilities, while MMMU is designed for multi-discipline evaluation. Considering our limited resources, we select a subset of MMT-Bench as MMT-S, which contains 83 tasks (19 categories). We use the development set and validation set of MMMU.

**Settings of prompt optimization** We evaluate our method in two settings: optimizing the whole prompt or optimizing the newly added phrase. MMT-Bench follows the most prevalent MLLM benchmark format, which uses the same prompt template within a task (e.g., `How many <object> are there in the image?` for the task of object counting). Thus, we optimize the whole prompt for each task in MMT-S. In MMMU, each question is identical, and thus we add an initialized meaningless phrase `Answer the questions about {task_name}` as the prompt to be optimized and move the original prompt to `<QUESTION>` in the meta prompt.

**Implementation details.** For MMT-S, we utilize the officially designated validation set as $\mathcal{D}_{few}$, which comprises approximately 10% of the total data, with roughly 20 samples per task. For MMMU, we combine the development and validation sets and allocate half of the data as $\mathcal{D}_{few}$. We follow VLMEvalKit by Duan et al. (2024) to implement the answer extraction module in $\mathcal{M}_A$. The total optimization iteration $N = 16$, with each round generating three new prompts. In each iteration, we select the top eight (i.e., $K = 8$) prompts for the meta prompt. We set the temperature to 1.0 when generating new prompts. During the optimization phase, we set $\alpha$ to 0.8 to encourage

the exploration of prompts that yield higher accuracy. In the final step, we set $\alpha^*$ to 0.6 to select the optimal prompt.

## 5.2 MAIN RESULTS

### 5.2.1 PERFORMANCE ANALYSIS

| Model | Original Score | TP-Eval Score | #Improved Task | Ratio |
|---|---|---|---|---|
| LLaVA-1.5-7B | 50.4 | 54.4 | 32 | 25.1% |
| DeepSeek-VL-7B | 55.2 | 57.3 | 21 | 23.3% |
| Mini-InternVL-Chat-4B-V1-5 | 54.6 | 56.9 | 16 | 40.4% |

Table 2: Overall result for MMT-S. All three models exhibited significant performance improvements across a substantial number of tasks following prompt customization.

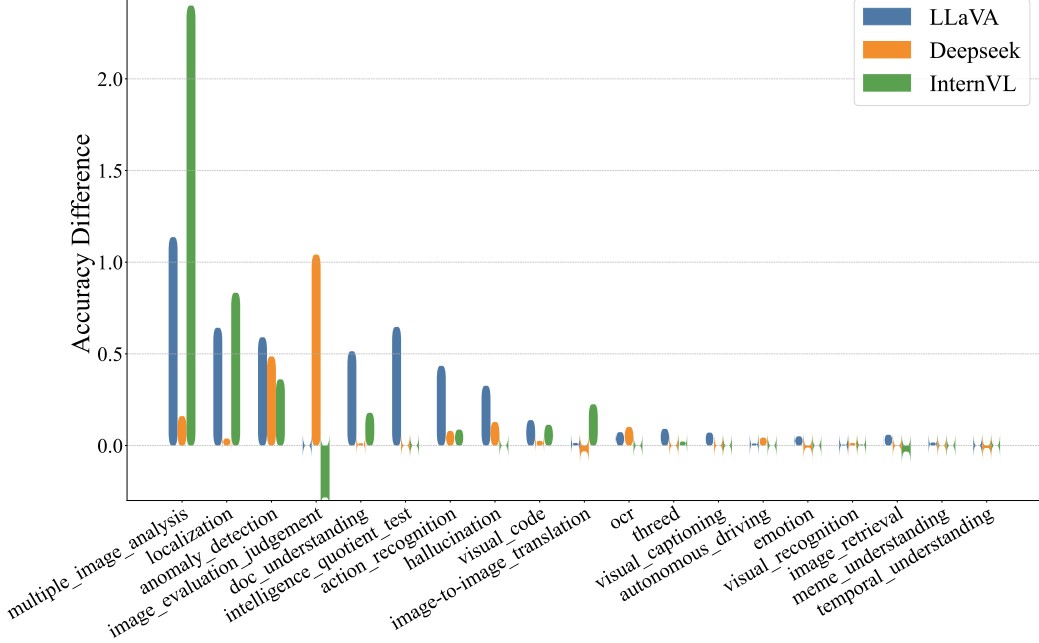

Figure 3: Results of different models on MMT-S (L2-category). Three models showed varying improvement across different task types while performance gains differ between models, highlighting the underestimation and bias introduced by original prompts and the effectiveness of our method.

**MMT-Bench** Table 2 shows the overall results of three open-source models' re-evaluated performance after prompt customization on MMT-S, where they exhibit varying degrees of improvement and show hidden potential across different tasks. It was observed that 32 tasks could yield a performance enhancement of 25.1% through prompt customization on LLaVA, ultimately leading to an overall score improvement of 4%. With respect to DeepSeek and InternVL, the former demonstrated a pronounced instruction-following capability during the experiments, while the latter exhibited a tendency towards detailed problem analysis. These characteristics render both models more robust to prompt changes, resulting in less accuracy improvement. The varying improvements suggest that models having similar scores may experience shifts in their rankings when the prompt changes. It also proves that prompt design flaws generally exist in MMT-Bench, resulting in a substantial underestimation of models' performance, while our evaluation method can tap their potential.

Fig. 3 shows more detailed L2-category level results of MMT-S. All three models did not demonstrate significant improvements in relatively simple and easily comprehensible tasks like visual

recognition, as well as in more challenging and complex tasks such as image retrieval. This outcome is comprehensive since, for the former, the model completely understands what it should do, and designing a more detailed prompt doesn't help; for the latter, model performance is mainly constrained by its inner true ability rather than inadequate prompts. Furthermore, certain tasks, such as anomaly detection, have been proven improvements across all three models, suggesting its general prompt design flaws. In other tasks like multiple image analysis and localization tasks, models show obvious otherness, where LLaVA and InternVL's performances demonstrate significant enhancements, but DeepSeek's barely maintains. This also emphasizes the validity of our proposed TP-Eval framework in mitigating bias and comparing model capabilities while ensuring fairness.

**MMMU** We conducted a comprehensive comparison of the results for all 30 tasks in MMMU. We evaluated the performance with the original questions, with the addition of the initial prefix prompt, and with the optimized prefix prompt on LLaVA. The results and improvements are summarized in Fig. 4.

It is evident that even adding the domain-specific initial prefix prompt (i.e., task name) can effectively guide the model to focus on and respond within that specific domain, thereby mitigating underestimation, but they are still too simple and not optimal. Compared to the initial prefix prompt, our optimized prefix prompts showed general performance improvements. Additionally, due to the semantic similarity metric in scores and extensive question information incorporated in the meta-prompt, the optimizer successfully generates prefix prompts with higher human readability and strong generalization capabilities within the domain.

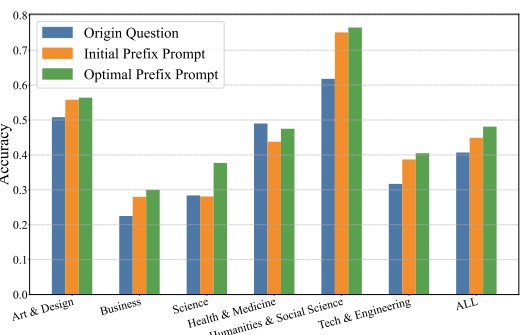

Figure 4: Overall performance with different prompt methods on MMMU with LLaVA. In most cases, the results after optimization surpass those achieved with the initial prompts, and they generally outperform the original questions as well.

Figure 5: Result of applying optimized prompts to other models. Applying customized prompts from one model to another yields performance changes that differ from each model's inherent characteristics.

### 5.2.2 OPTIMALITY ANALYSIS

Fig. 5 presents the overall results obtained from a hybridization of customization outcomes across different models within MMT-S. It is evident that prompts optimized using a model itself as a scorer yield superior performance. Notably, when prompts optimized on InternVL are applied to LLaVA and DeepSeek-VL, their performance will decline. This outcome not only supports that the optimal prompts proposed for one specific model may not be universally applicable, thereby underscoring the necessity of customizing prompts to tap the models' full potential but also indicates that our method has indeed approached the objective of customization and can effectively support TP-Eval framework.

### 5.2.3 ERROR ANALYSIS

Similar to many text-only prompt optimizations, our method, while ensuring an overall enhancement in performance and a closer alignment with the true capability for evaluated models, may still encounter optimization failures for a limited number of tasks. This can, in turn, result in a slight

performance deterioration when using optimized prompts. For instance, although LLaVA has an overall improvement of 25% across 32 tasks, it also experiences an approximate 6% performance decline on 6 tasks. We argue that a critical factor contributing to this is the relatively small size of the validation set currently designated by the official multi-modal benchmarks, which may cause overfitting on the training set. Despite our efforts to incorporate introspection mechanisms for more effective utilization of few-shot data, and the implementation of re-ranking and meta-prompt design strategies to mitigate overfitting, this challenge persists, but its impact remains relatively minor.

## 5.3 ABLATION STUDY

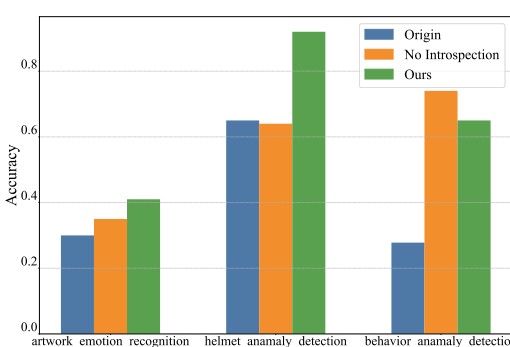

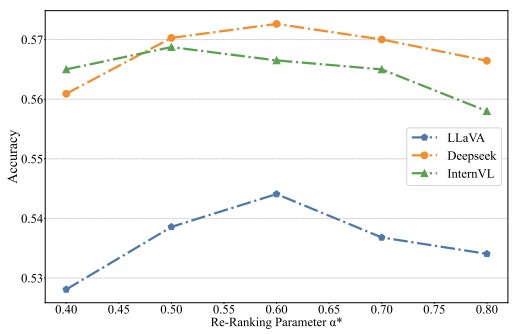

Figure 6: Performance on whether to use introspection or not.

Figure 7: Influence of re-ranking. Both excessively high and low $\alpha$ can lead to a reduction in performance, and each model achieves optimal performance with $\alpha \in [0.5, 0.6]$.

**Introspection** Fig. 6 illustrates the results of LLaVA in three tasks of MMT-S when introspection is not incorporated. It is evident that the optimization results on both artwork emotion recognition and helmet anomaly detection tasks are significantly inferior to those achieved with our method. Notably, the latter even experiences a failure in optimization, resulting in a performance decline. This underscores the effectiveness of integrating introspection to enhance the few-shot optimization capability on multi-modal benchmarks. Furthermore, the figure indicates that the accuracy of behavior anomaly detection is better without introspection. This phenomenon arises from the prompt explicitly designating choice A as normal and choice B as abnormal, disregarding the randomized initial order of the choices presented in this task. This is an instance of semantic overfitting that leads to misleadingly high performance. Thus, the introduction of introspection can also enhance result interpretability.

**Re-ranking parameter.** Fig. 7 illustrates the impact of varying the proportion of accuracy during the re-ranking phase on optimization results. As depicted, when setting the parameter to 0.8, which in fact omits the re-ranking stage, leads to significant overfitting and ultimately degrades the optimization outcomes. Conversely, a disproportionately low correctness ratio may result in the exclusion of potentially optimal prompts, thereby underfitting and hindering the optimized prompts from fully leveraging the model's capabilities. Based on our experiments, we conclude that a value between 0.5 and 0.6 is appropriate to ensure both effectiveness and coherence across the models.

## 5.4 ZERO-SHOT EXPLORATION

Considering that the task may suffer from extremely limited data availability or involve privacy concerns that prevent the disclosure of answers, it becomes impractical to construct even one training sample. In response, we propose an approach that leverages the robust In-Context Learning(ICL) capabilities of LLMs to extend our method to the zero-shot scenario. Specifically, we aim to optimize prompts for a newly introduced task through the use of a selection of previously successfully optimized examples, thereby facilitating zero-shot customizing. We anticipate that LLMs can discern certain vocabulary, phrases, and reasoning patterns that the model under examination may prefer from these ICL examples. A straightforward experiment result illustrating this observation can be found in Table 3, where we select 3 tasks from all 32 MMT-S underestimated tasks for LLaVA as targets and use the rest as ICL examples. We use this zero-shot ICL-based optimization fashion to

refine the original prompts, which also enhances the original accuracy and is close to that of optimal prompts learned by 20 examples.

| Task name | Original prompt | Zero-shot | Optimal prompt |
|---|---|---|---|
| helmet anomaly detection | 0.65 | 0.86 | 0.92 |
| artwork emotion recognition | 0.3 | 0.33 | 0.41 |
| spot similarity | 0.23 | 0.42 | 0.52 |

Table 3: Zero-shot prompt optimization utilizing In-context Learning.

## 6 CONCLUSION

We investigated MLLM benchmarks and found that overly simplistic or unsuitable textual prompts may lead to an underestimation of models' capabilities. To address this issue, we propose an ideal evaluation framework, TP-Eval, which customizes the most suitable task prompt for each model to mitigate prompt-induced biases and tap the models' potential. To achieve this goal, we drew on the successful experiences of automatic prompt optimization on text-only LLMs and designed a prompt optimization method tailored to the few-shot scenario of MLLM benchmarks. Our experiment results for three models on the MMT and MMMU indicate the effectiveness of our method.

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
