# OpenReview forum: "TP-Eval: Tap Multimodal LLMs' Potential in Evaluation by Customizing Prompts"
_ICLR.cc/2025/Conference — Submitted to ICLR 2025_

### Official Review · Reviewer_XiKc · 2024-11-02

**Soundness:** 3
**Presentation:** 3
**Contribution:** 2
**Rating:** 5
**Confidence:** 2

**Summary:**

The paper addresses the issue of prompt sensitivity in the evaluation of multimodal large language models (MLLMs). It argues that current benchmarks overlook the impact of prompt variations on model performance, which can lead to significant fluctuations and underestimation of capabilities. The paper introduces a new evaluation framework named TP-Eval, which customizes prompts for different models to reduce evaluation biases and tap into the models' potential more effectively.

**Strengths:**

1. The paper analyzes and highlights the problems with existing MLLM benchmarks that lead to underestimation and evaluation bias due to prompt sensitivity.
2. It introduces TP-Eval, an evaluation framework that customizes optimal prompts for different models through automatic prompt optimization, tailored to MLLM benchmarks.

**Weaknesses:**

1. I think the authors should deeply analyze and summarize prompt improvement based on TP-Eval and provide more insights for building reasonable benchmarks in the future.

**Questions:**

I have a question about the motivation of the paper. Why optimize the prompt for the MLLM benchmark? I think this paper proposes a good method for automation prompt engineering. I also understand the problem of prompt sensitivity. A slight change in the prompt will cause a drastic change in the evaluation results.

However prompt sensitivity should be a problem with the MLLM itself. The more robust the model, the more stable the evaluation results. In addition, the same evaluation set can ensure the fairness of the evaluation. Therefore, the optimization direction should be on MLLM, not the evaluation benchmark.

---

> ### Author Response · Authors · 2024-11-24
>
> We sincerely thank reviewer XiKc for the valuable comments. We give detailed responses to address your concerns as follows.
>
> **Weakness: Lack of systematic analysis and summary about prompt improvement, lack of more insights for ideal benchmark design.**
>
> Response: Thank you for your comment. Due to the serious prompt sensitivity and preference among different models we utilized in our research, the prompt optimization direction is irregular and incomprehensible to humans. As an example shown in *Table 1*, minor modifications to the prompt resulted in significant performance variations across the two models. Another example in  *Figure 1(a)*, after we only removed the sentence "Focus on visual cues", we get a a significant drop in performance. But the sentence itself doesn't matter. **Hence, it is difficult for humans to analyze and summarize reasonable patterns directly**.
>
> Although it's hard to manually summarize universal principles, in *Section 5.4*, we explored the use of the in-context learning capabilities of LLMs to infer potential prompt optimization patterns from optimized prompts of previous tasks and directly generate better ones. Experiments have demonstrated the feasibility of this approach of **replacing humans with advanced LLMs for summarizing and suggesting**, and we will continue to follow up on this in further research.
>
> **Question: Prompt sensitivity should be a problem with the MLLM itself. The optimization direction should be on MLLM, not the evaluation benchmark.**
>
>
> Response: We fully agree with your suggestion that improving the models themselves is important, and at the same time, we believe that **eliminating prompt sensitivity for MLLMs in evaluation can find out where the model really needs to be improved benefits the development of models**, which is important as well. In our early experiments, which are not demonstrated in the paper, even the SOTA MLLMs like GPT-4o, can still have a performance difference of up to 3% before and after prompt customization on the part of MMT-Bench. This indicates that prompt sensitivity is a common and unsolvable problem for MLLMs, which strongly proves our work's value in eliminating sensitivity by prompt customization.
> As we have emphasized in *Section 2.1*, **summarization of optimization patterns for future benchmark design cannot fundamentally solve this issue**, that's because current benchmark designers tend to use homogeneous prompts, but different models have different preferences for prompts, hence using the same prompt inevitably introduces bias. Benchmark designers, whether intentionally or unintentionally, **may favor or deviate from the preferences of certain models**, which can cause significant performance divergence from their true capabilities (as analyzed in detail in *Section 2*). Thus, **we cannot determine whether the performance deficiency is caused by prompt sensitivity or the model's inherent capability**. Unfortunately, current designers don't realize this prompt sensitivity unfairness but only concentrate on designing new benchmarks. That's why we propose the TP-Eval framework, which 'decouples' the model's inherent capability and its prompt generalization ability, effectively excluding the influence of prompt sensitivity, making the benchmark more objective in order to help provide valuable insights to **guide model iterations**.
>
> This issue requires essential long-term efforts, and our TP-Eval framework provides a practical solution to the current challenges. We hope our work can raise awareness about the prompt sensitivity problem in MLLM evaluation. Meanwhile, We will continue to follow up on this in our future research.

---

> > ### Comment · Reviewer_XiKc · 2024-11-26
> > **Response to author**
> >
> > Thank you for your answers to my questions. However, my concern still remains.
> >
> > You mentioned “different models have different preferences for prompts, hence using the same prompt inevitably introduces bias”, which is very much worth discussing. In practical applications, users cannot cater to the preferences of the model when prompting it. Additionally, recording "prompt optimization direction is irregular and incomprehensible to humans. ", how should users utilize the model in practical applications to achieve the best performance?

---

> > > ### Author Response · Authors · 2024-11-27
> > >
> > > We sincerely thank reviewer XiKc for the valuable reply. We would like to make some clarification.
> > >
> > >
> > > Our framework mainly focuses on MLLM evaluation scenarios, which are somewhat different from daily-use scenarios. Even though we focus on evaluation more, this is not without relation to daily use. Since we use GPT to generate candidate prompts, **the optimized prompts are themselves comprehensible to humans and may be input in daily-use scenarios by humans**. For example, *Is the person in the picture wearing a helmet?* is optimized as *Please analyze the image for head protection elements. Focus on clear visual indicators to ensure accurate assessment.* Here it uses a broader term *head protection elements* instead of *helmet* to increase generalization, and the addition of a reminder *focus*, are changes that a human being would fully understand and input.
> > >
> > >
> > > However, as I mentioned before, **the direction of optimization is usually irregular**, as small semantically meaningless changes in human perception may significantly bias the model performance, and such prompt sensitivities and preferences are strongly model-dependent, hence the pattern of improvement cannot be summarised by humans. Our zero-shot ICL experiments in *Section 5.4*, on the other hand, show that powerful **LLMs are significantly better at summarising patterns** here than humans, pointing to a possible direction of development.
> > >
> > >
> > > An important purpose of the evaluation is to drive the improvement of the capabilities of MLLMs, which benefits the human experience of daily use. Under the existing evaluation benchmarks, our TP-Eval framework decouples the model's inherent capability and prompt generalization capability, **allowing us to better identify model deficiencies, guiding effective improvements accurately, and finally, improving the user experience.** We believe that by decoupling the prompt sensitivity when evaluating, we can obtain more precise guidance to enhance the prompt generalization capability. Thus, **users can avoid the obstacle of repeatedly modifying prompts in their daily use**.

---

> ### Author Response · Authors · 2024-11-25
>
> Dear Reviewer XiKc,
>
> Thanks very much for your thorough review and valuable comments. As the deadline for the Author/Reviewer discussion is approaching, it would be nice of you to let us know whether our answers have solved your concerns. We are happy to provide any additional clarifications that you may need.
>
> Best regards,
>
> Authors

---

> ### Author Response · Authors · 2024-11-29
>
> Dear Reviewer XiKc,
>
> We appreciate your attention and response to our rebuttals! As the discussion period is nearing its conclusion, we kindly ask if you could review our response to ensure it addresses your concerns. Your feedback is greatly appreciated.
>
> Thank you for your time!
>
> Best,
>
> Authors

---

> > ### Author Response · Authors · 2024-12-02
> >
> > Dear Reviewer XiKc,
> >
> > We appreciate your attention and response to our rebuttals! As the discussion period is **nearing its conclusion**, we kindly ask if you could review our response to ensure it addresses your concerns. **Your feedback is greatly appreciated**.
> >
> > Thank you for your time!
> >
> > Best,
> >
> > Authors

---

### Official Review · Reviewer_xbei · 2024-11-03

**Soundness:** 2
**Presentation:** 2
**Contribution:** 2
**Rating:** 3
**Confidence:** 5

**Summary:**

The paper tries to addresse the challenge of evaluating MLLMs accurately. The authors highlight that current benchmarks often overlook the sensitivity of MLLMs to prompt variations, which can lead to significant performance fluctuations and underestimation of model capabilities. They propose a new evaluation framework called TP-Eval, which aims to reduce evaluation biases and unlock the potential of MLLMs by customizing prompts for each model.
TP-Eval introduces a framework includes several modules for prompt customization, taking into account the unique scenario of MLLM evaluation. The authors conduct extensive experiments to demonstrate the effectiveness of TP-Eval in uncovering the true capabilities of MLLMs. They show that by rewriting original prompts to customized prompts for different models, TP-Eval can significantly improve the performance of MLLMs across various tasks.
The authors argue that the true capabilities of MLLMs should be assessed under optimal prompts, which are derived from slight modifications of the original prompts while maintaining the semantic integrity of the task instructions. TP-Eval ensures fairness across models and manages labor costs by automating the prompt customization process.

**Strengths:**

The paper proposes a novel evaluation framework, TP-Eval, which addresses the issue of prompt sensitivity in MLLMs. This is an innovative approach to improving the accuracy and reliability of MLLM evaluations.

**Weaknesses:**

This approach may raise concerns about over-optimization for the test set. MLLMs are capable of understanding a wider range of prompts and following instructions to provide responses, which is an issue that needs to be addressed during the alignment phase using methods such as SFT or RL. Existing evaluations primarily assess the capabilities of MLLMs during the IFT or pre-training stages.
The experiments are not sufficient, failing to consider the consistency at different stages of MLLMs, such as pretrain/IFT/alignment models, nor do they take into account the uncertainties and instabilities brought by personalized prompts to model iterations.

**Questions:**

1. In user usage scenarios, optimized prompts are unlikely to be used, instead, ‘poor’ or ‘average’ prompts are more probable. Therefore, does the underestimation of MLLMs mentioned in the article actually exist? Is there a suspicion of over-optimization for the test set?
2. TP-Eval represents the performance of each model under their respective best prompts. How correlated is this with the effect of randomly selecting an "average" prompt? It would be appreciated if corresponding conclusions could be supplemented.
3. Pretrained or IFT MLLMs can mostly only follow simple instructions. Can TP-Eval provide corresponding experimental results on such models?
4. How to use this benchmark for model iteration? For example, if change to a more powerful vision encoder or LLM backbone, will it be reflected in the results?

---

> ### Author Response · Authors · 2024-11-24
>
> We sincerely thank reviewer xbei for the valuable comments. We give detailed responses to address your concerns as follows.
>
> **Weakness: Missing experiments on different stages of MLLMs. Personalized prompts bring uncertainties and instabilities to model iterations.**
>
> Response: Thank you for your comment. Employing our evaluation framework can better identify the bottleneck and benefit model iteration.
> An important objective of MLLM evaluation is to discover the strengths and weaknesses of the model to guide model iterations. Existing benchmarks use a unified prompt for all models, but each model exhibits different prompt sensitivity and preference. **Existing benchmarks couple prompt sensitivity and inherent capabilities in the evaluation, which may bias results and conclusions and misguide the model iteration**. For example, if a model performs poorly on a task, is it caused by inherent capabilities or prompt sensitivity of this task? It leads to different plans for model iteration.
>
> The TP-Eval employs prompt customization for each model and thus **decouples a model's inherent capabilities from its prompt generalization ability in the evaluation** on the instance or task level. If the differences between using customization and not are large, the bottleneck should be prompt generalization ability, and we can do task-related instruction augmentation on the IFT. Conversely, if the differences are minor, the bottleneck should be inherent capabilities, and we can collect more diverse related data, especially in the early training stage. Thus, the customization framework helps assess models more accurately and provides valuable insights to **guide model iterations**.
>
> We follow existing MLLM evaluation benchmarks, such as MMMU, MMT-Bench, and MMBench, to evaluate the final stage of MLLMs (released version). We will release our code, and researchers can do stage-wise evaluations of their models to assist their model iterations more precisely.

---

> ### Author Response · Authors · 2024-11-24
>
> **Question 1: Does the underestimation of MLLMs mentioned in the article actually exist? Is there a suspicion of over-optimization for the test set?**
>
> Response: We focus on evaluating model performance in specialized, professional scenarios rather than daily usage. The meaning of professional evaluation is accurately assessing a model's abilities, finding its weaknesses, and guiding model iteration. And TP-Eval can enhance existing professional evaluation.
>
> But it doesn't mean that TP-Eval is totally separated from daily usage. In daily life, people tend to use direct and simple prompts, similar to benchmarks, and may not fully utilize the model's potential. Such prompts will sometimes be unreliable, which we have shown in our paper. Moreover, when people don't get satisfactory responses from models, they also tend to design a better prompt, which TP-Eval actually does automatically.
>
>
>
> Through our experiments with three models and two different benchmarks, we have proved that underestimation actually exists. For example, LLaVA-v1.5-7b has a performance improvement of 4 percent on MMT-Bench after customization prompts, which is approximately the performance gap between LLaVA-v1.5-7b and LLaVA-v1.5-13b. The example in *Figure 1(a)* shows that the model can't understand the prompt in the benchmark, which causes underestimation. More detailed experiment results can be seen in *Sections 5.1 and 5.2*. Moreover, our early research also shows that even SOTA models like GPT-4o have been underestimated. Due to the cost of calling API, we didn't test on the whole MMT-Bench.
>
>
>
> **Question 2: Relation between average and optimal prompts**
>
> Response: For the diversity of natural language, it is impractical to enumerate all possible prompts and randomly select an 'average' prompt. However, through extensive experiments, we observed that the performance of the original simple prompt in benchmarks is generally positioned around the median of the limited set of alternative prompts we tested. Therefore, we can regard the initial prompt as approximating an 'average' prompt. According to the experimental results in *Section 5.2.1*, the optimized prompts consistently outperformed or, at worst, matched the performance of the original prompts, which also demonstrates the effectiveness of our method.
>
>
>
> **Question 3: Experiment on Pretrained or IFT MLLMs**
>
> Response: Our study focuses on prompt customization for the evaluation of the finished model. Consequently, a comparison of models switched to different training stages was not performed. However, as previously stated in weakness, the application of the TP-Eval framework to the iterative process, whereby the inherent capability and prompt generalization capability are decoupled, will prove an effective means of identifying the areas in which the model requires improvement. The difficulty of locating open-source models with disparate training phases on the internet has failed us to furnish a comprehensive and detailed experimental report at this time. However, the TP-Eval source code will be made available to the community in the near future, allowing its application to model iterations. Furthermore, we will endeavor to provide the experimental data if it is deemed necessary.
>
> **Question 4: Utilization on iteration**
>
> Response: Our focus is on prompt customization for the evaluation of the finished model, and thus we did not perform a comparison of models switched to different architectures. However, as we mentioned in the weakness, applying the TP-Eval framework into the iterative process by decoupling the inherent capability and the prompt generalization capability will be effective in finding out where the model really needs to be improved. Due to the difficulty of finding open-source models in training stages on the web, we are unable to provide you with a detailed and comprehensive experimental report right away. However, we will soon open-source the TP-Eval source code to the community so that you can apply it to your model iterations, and we will endeavor to provide you with the experimental data if you think it is necessary.

---

> ### Author Response · Authors · 2024-11-25
>
> Dear Reviewer xbei,
>
> Thank you for the precious review time and insightful comments on our paper. We have provided corresponding responses to address your concerns in detail. If you have any other concerns, we are more than happy to provide additional clarification at any time. Sincerely looking forward to your reply!
>
> Best regards,
>
> Authors

---

> ### Author Response · Authors · 2024-11-29
>
> Dear Reviewer xbei,
>
> As the discussion period is nearing its conclusion, we kindly ask if you could review our response to ensure it addresses your concerns. Your feedback is greatly appreciated. Please contact us as soon as possible if you need experimental data that you are interested in.
>
> Thank you for your time!
>
> Best,
>
> Authors

---

> > ### Author Response · Authors · 2024-12-02
> >
> > Dear Reviewer xbei,
> >
> > As the discussion period is **nearing its conclusion**, we kindly ask if you could review our response to ensure it addresses your concerns. **Your feedback is greatly appreciated**. Please contact us as soon as possible if you need experimental data that you are interested in.
> >
> > Thank you for your time!
> >
> > Best,
> >
> > Authors

---

### Official Review · Reviewer_8WzZ · 2024-11-04

**Soundness:** 3
**Presentation:** 3
**Contribution:** 2
**Rating:** 6
**Confidence:** 4

**Summary:**

This paper introduces an automated prompt optimization framework, TP-Eval, which addresses the prompt sensitivity problem in multimodal large language models.

**Strengths:**

Strength
1.Innovative Approach (Prompt Customization): This paper introduces an automated prompt optimization framework, TP-Eval, which addresses the prompt sensitivity problem in multimodal large language models. By customizing prompts with a small sample set, this method effectively reduces prompt bias, making an innovative contribution to the evaluation of multimodal models.
2.Introduction of an Introspection Mechanism: The introspection mechanism aids in prompt refinement during optimization, not only improving the prompts but also avoiding simple semantic drift. This mechanism enhances model accuracy across diverse tasks, demonstrating a practical and effective design for improved performance on complex tasks.
3.Exploration of Zero-Shot Optimization for Data-Limited Scenarios: The zero-shot optimization method successfully addresses data scarcity or privacy-restricted scenarios through In-Context Learning (ICL) examples. Experimental results show that, even without samples, the model achieves optimized performance through zero-shot methods, representing a practical extension of TP-Eval.
4.Comprehensive Experimental Design: The paper’s experiments cover various task types and include ablation studies and error analysis, providing a thorough view of TP-Eval’s strengths and limitations with well-supported data. These experiments offer reviewers a solid basis for understanding the method’s effectiveness.
5.Revealing Previously Inflated Metrics with Introspection Mechanism: The introspection mechanism reveals that certain high metrics previously observed in some tasks were inflated due to the model merely “memorizing” specific prompt patterns rather than genuinely understanding task requirements. For instance, in anomaly detection tasks, explicit prompt instructions identifying option A as normal and B as abnormal led the model to rely on option order, resulting in semantic overfitting. The introspection mechanism breaks this rigid pattern, requiring the model to “understand” the task more deeply, thereby exposing misleading high scores previously masked. (Section 5.3, Ablation Study)

**Weaknesses:**

1.Overfitting Risk Not Fully Addressed: Although the paper introduces introspection and reordering mechanisms, some tasks in the experiments still experience optimization failures or overfitting (as seen in error analysis, Section 5.2.3). This issue may be more pronounced in multimodal tasks with smaller datasets. The robustness and generalizability of TP-Eval require further validation.
2.Strong Model Dependency: Experimental results (Figure 5) show that different models respond quite differently to the same optimized prompts, indicating that TP-Eval requires individual tuning and optimization for each model, lacking general applicability. This raises costs for multi-model or large-scale deployments, limiting the method's practicality.

**Questions:**

I don't have questions.

---

> ### Author Response · Authors · 2024-11-24
>
> We sincerely thank reviewer 8WzZ for recognizing our significant insight and contribution to the MLLM evaluation. We give detailed responses to address your concerns as follows.
>
> **Weakness 1: Overfitting Risk Not Fully Addressed**
>
> Response: Thank you for your comment. As described in the paper, we also highlighted that overfitting risk is one of the main challenges of prompt customization. Meanwhile, the state-of-the-art prompt optimization method, OPRO, also struggles to address this issue fully. Thus, we introduce well-designed modules, such as self-introspection, editing distance, and generating temperature, to significantly alleviate overfitting risk. As mentioned in experiment *Section 5.3*, self-introspection significantly alleviates this risk.
> Besides, we do the experiments you mentioned about fewer samples. Specifically, on randomly selected five tasks in MMT-Bench, LLaVA-1.5 achieved near-average accuracy of 0.57, 0.59, and 0.7 using 5, 10, and 20 samples, respectively (details are demonstrated in the table below). Using fewer samples will definitely decline optimization results, but **overfitting doesn't appear when using only five samples**, demonstrating overfitting is not a significant problem for our method on smaller datasets.
>
> | Task                        | Original Prompt | 5 examples   | 10  examples | 20 examples  |
> | --------------------------- | ---------- | ---- | ---- | ---- |
> | Artwork Emotion Recognition | 0.30 | 0.36 | 0.33 | 0.41 |
> | Spot the Similarity         | 0.23 | 0.47 | 0.49 | 0.57 |
> | Spot the Diff               | 0.83 | 0.93 | 0.90 | 0.95 |
> | Behavior Anomaly Detection  | 0.28 | 0.43 | 0.59 | 0.65 |
> | Helmet Anomaly Detection    | 0.65 | 0.65 | 0.65 | 0.92 |
>
> Moreover, according to the experimental results in *Section 5.4*, our method exhibits **strong zero-shot transferring generalization ability**.
>
> **Weakness 2: Strong Model Dependency**
>
> Response: Thank you for your comment. We have to claim that model dependency is not a weakness of our method. Instead, it is one of our motivations and the inherent attribute of prompt customization. Since current MLLMs show distinct prompt sensitivity and preference, we must customize model-dependent prompts for different models. Besides, our method is not a general prompt optimization method to fight for better accuracy. Our objective is to disentangle the effect of prompt sensitivity in MLLM evaluation for better analysis to find the real bottleneck of the model. Thus, it must be model-dependent.

---

> ### Author Response · Authors · 2024-11-25
>
> Dear Reviewer 8WzZ,
>
> We express gratitude for your time spent reviewing and your valuable comments. We have provided responses to address your concerns. We look forward to engaging in further discussion to confirm whether or not your concerns have been addressed.
>
> Best regards,
>
> Authors

---

> ### Author Response · Authors · 2024-11-29
>
> Dear Reviewer 8WzZ,
>
> As the discussion period is nearing its conclusion, we kindly ask if you could review our response to ensure it addresses your concerns. Your feedback is greatly appreciated.
>
> Thank you for your time!
>
> Best,
>
> Authors

---

> > ### Author Response · Authors · 2024-12-02
> >
> > Dear Reviewer 8WzZ,
> >
> > As the discussion period is **nearing its conclusion**, we kindly ask if you could review our response to ensure it addresses your concerns. **Your feedback is greatly appreciated**.
> >
> > Thank you for your time!
> >
> > Best,
> >
> > Authors

---

### Meta-Review · Area_Chair_zRbQ · 2024-12-18

**Metareview:**

This paper introduces the interesting idea that, due to prompt sensitivity, LLMs should be evaluated using a prompt that is optimal for that LLMs for each task.  Specifically they use GPT4o to create a collection of prompts for each LLM for each task and give the LLMs best score on the task.

Overall the reviewers did not seem to think that this was a very interesting paper.  One reviewer questioned the authors motivation for doing this because although it gives a different insight, the results of the optimized prompts seem to be in line with the non-optimized prompts and as another reviewer points out finding the optimal prompt for every model for every question is expensive.

The reviewers raise concerns of overfitting, the possibility that the model will learn from the prompt optimization step and a lack of sufficient experiments.  The authors refute these in a reasonable way, but the main issue, that there seems to be no strong motivation for this costly framework, remains.  In a future submission please show how using this framework would impact the relative performance of one model versus another.  If the relative performance remains the same, it seems that the cost of using this framework is not justified.  This paper needs to be re-written with a stronger motivation for the framework.

**Additional Comments On Reviewer Discussion:**

Unfortunately, there was not any response from the reviewers to the authors rebuttals.  I believe that this is because the reviewers were not enthusiastic about the paper and seemed disinterested in the authors responses.  Even the most positive reviewer (6) stated that they had no questions for the authors and that the method was subject to overfitting which they seemed to be convinced about and cited the authors own results as evidence.  The strongest negative reviewer (3) had absolute confidence in their assessment and did not respond to authors rebuttal.  The slightly negative

---

### Decision · Program_Chairs · 2025-01-22

Reject